# LANE: Label-Aware Noise Elimination for Fine-Grained Text Classification

## Abstract

In this paper, we propose Label-Aware Noise Elimination (LANE), a new approach that improves the robustness of deep learning models when trained under increased label noise in fine-grained text classification. LANE leverages the semantic relations between classes and monitors the training dynamics of the model on each training example to dynamically lower the importance of training examples that are perceived to have noisy labels. We test the effectiveness of LANE in fine-grained text classification and benchmark our approach on a wide variety of datasets with various number of classes and various amounts of label noise. LANE considerably outperforms strong baselines on all datasets, obtaining significant improvements ranging from an average improvement of $2.4\%$ in F1 on manually annotated datasets to a considerable average improvement of $4.5\%$ F1 on datasets with higher levels of label noise. We carry out comprehensive analyses of LANE and identify the key components that lead to its success.

## 1 Introduction

Deep learning models are increasingly powerful in many NLP applications, but their success is often hindered by data quality. Many existing datasets are annotated by humans on crowdsourcing platforms Demszky et al. (2020) or by automatic approaches such as distant (or weak) supervision Mintz et al. (2009); Wang et al. (2012); Abdul-Mageed & Ungar (2017), and, while weak supervision inherently introduces unwanted mislabeled examples, humans—no matter how careful, are also prone to making labeling errors especially on fine-grained tasks that involve distinguishing between a large number of closely confusable or overlapping classes, e.g., emotion detection Mohammad (2012); Islam et al. (2019); Bao et al. (2009); Strapparava et al. (2012); Liu et al. (2019) or fine-grained topic classification tasks Lewis et al. (2004). The mislabeled training examples are particularly harmful when learning large overparameterized neural networks, since these networks can achieve zero training error on any dataset, with very poor generalization capabilities Zhang et al. (2016).

Several works Li et al. (2023); Karim et al. (2022); Liu & Guo (2020) designed various changes to the training process to learn under label noise. For example, Peer Loss Function Liu & Guo (2020) alters the training loss function to account for label noise, DISC Li et al. (2023) utilizes an instance-specific dynamic thresholding mechanism that blocks access to specific training examples based on the momentum of each instance's memorization strength. Unicon Karim et al. (2022) leverages a semi-supervised learning (SSL) framework that considers potentially noisy labeled data as unlabeled examples in an SSL algorithm. Area Under the Margin (AUM) Pleiss et al. (2020) utilizes an instance-specific average margin that identifies potentially mislabeled examples from the training set according to the model's behavior on these examples and blocks access to these examples through a fixed threshold. AUM measures the average difference between the logit values corresponding to a sample's *assigned* label and its largest *non-assigned* label calculated across the training epochs. The AUM for a *mislabeled sample* is expected to be low, likely negative since the model—through generalization from other correctly labeled training samples, tends to predict the sample in its (hidden) true class which is different from the (wrongly) assigned class, and hence, the largest logit (among all logits) no longer corresponds to the assigned (wrong) label Pleiss et al. (2020). After this data characterization by AUM, Pleiss et al. (2020) subsequently remove samples with low AUM from the training set using a fixed rigid AUM threshold (i.e., the $95$ percentile).

However, we posit that, through this fixed threshold used to remove mislabeled samples, difficult but valuable samples that exist under the threshold are unnecessarily removed from the training set. In addition, the computation of AUM that contrasts two labels (the assigned—potentially wrong—label and the largest non-assigned label) treats labels independently, and thus, ignores semantic similarities that inherently exist between fine-grained classes (e.g., in fine-grained emotion detection tasks, "anger" is semantically more similar to "fear" than it is to "joy"). To this end, we introduce **L**abel-**A**ware **N**oise **E**limination (LANE), a novel approach that identifies mislabeled or noisy samples from the training data and seamlessly mitigates their harmful effects. Unlike Pleiss et al. (2020) who remove mislabeled or ambiguous samples from the training set using a fixed threshold, we improve the robstness of our model under label noise by retaining *all* training samples but re-weighting them differently based on the model's behavior on these samples measured against their assigned labels. In re-weighting the samples, we estimate the degree of "noisiness" of the assigned labels by introducing *label-aware margins* averaged across training iterations that capture inter-class semantic similarities. For example, a sample with true label "anger" but with assigned label "joy" is noisier (has a higher degree of noisiness) than a sample with true label "anger" but with assigned label "fear" since "fear" is semantically closer to "anger" than "joy". Our label-aware margins extend the concept of *margins* Pleiss et al. (2020) by adaptively weighting samples when the (hidden) true label and the (wrongly) assigned label do not match. Precisely, we capture inter-class semantic similarities and dynamically lower samples' weights if the model perceives them as noisy (the noisier the assigned label the lower the weight). We learn the inter-class semantic similarities using a label-aware supervised contrastive loss to improve the capabilities of the model to distinguish between easily confusable samples by bringing the latent representations of input samples closer together if they belong to semantically similar classes and further apart if they belong to semantically dissimilar classes.

We evaluate the effectiveness of LANE on multiple well-established fine-grained datasets: Empathetic Dialogues Rashkin et al. (2019), GoEmotions Demszky et al. (2020), ISEAR Scherer & Wallbott (1994), CancerEMO Sosea & Caragea (2020), RCV1 Lewis et al. (2004), SciHTC Sadat & Caragea (2022), SST-5 Socher et al. (2013a), Amazon Review McAuley & Leskovec (2013), Yelp Review Asghar (2016), and Yahoo Answer Chang et al. (2008). Using these datasets, we show that LANE works well on various tasks and domains (emotion and general text classification; social networks, dialogues, and personal experiences). In all our experiments, automatically scaling down the importance of identified noisy samples from the training set shows great potential, improving the overall performance on our original datasets by $2.4\%$ F1 on average over the strong AUM approach Pleiss et al. (2020) and by $4.5\%$ F1 on average on our datasets with higher levels of label noise.

We summarize our contributions as follows: **1)** We introduce LANE, a new approach that leverages inter-class semantic similarities and monitors the training dynamics of each training example to automatically identify and minimize the harmful effects of ambiguous or mislabeled examples; **2)** We evaluate the effectiveness of our approach on ten text classification benchmark datasets from different tasks and domains; **3)** We carry out a comprehensive analysis and ablation study of LANE and analyze how it performs on datasets that have different levels of noise.

## 2 RELATED WORK

Learning with label noise have started to received substantial attention due to the high risk of deep learning models to overfit Liu & Tao (2015); Goldberger & Ben-Reuven (2016); Ren et al. (2018); Englesson & Azizpour (2021); Zhang & Plank (2021); Margatina et al. (2021); Li et al. (2021); Plank (2022); Karim et al. (2022); Garg et al. (2023); Wei et al. (2023c;b;a). For example, Goldberger & Ben-Reuven (2016) propose adding a noise layer in the neural network architecture, whose parameters can be learned for an accurate label estimation. Saxena et al. (2019) introduce a curriculum-learning approach that uses learnable data parameters to rank the importance of examples in the learning process. These parameters are then leveraged to decide the data to use at different training stages. Liu & Guo (2020) on the other hand propose to alter the loss function to make it more robust in the face of label noise and introduce Peer Loss Functions, which evaluate predictions on both the samples at hand, as well as carefully automatically constructed *peer* samples. Other approaches focus on data quality and design techniques to accurately identify and eliminate potentially mislabeled instances Brodley & Friedl (1999); Pleiss et al. (2020); Swayamdipta et al. (2020). For example, Swayamdipta et al. (2020) introduce data cartography, a model-based tool that separates training data into three (potentially overlapping) regions, easy-to-learn, ambiguous, and hard-to-learn (many of

which are mislabeled), and re-trains on each data region to understand its benefits to learning and generalization. Pleiss et al. (2020) identify and subsequently remove mislabeled training samples by monitoring the behavior of the model on each sample and estimating its Area Under the Margin (AUM) to determine what to remove from the data. Our work builds on this approach: we reformulate the Area Under the Margin Pleiss et al. (2020) and leverage the inter-class semantic similarities present in fine-grained tasks to improve training data quality and diminishing the harmful effects of noisy samples by reweighting the importance of samples during training.

The idea of weighting each training example has been well studied in the literature. A classical method in statistics is importance sampling Kahn & Marshall (1953), which assigns weights to samples in order to align one distribution to another. Boosting algorithms such as AdaBoost Freund et al. (1999), select harder samples to train subsequent classifiers. Focal loss Lin et al. (2017) incorporates a soft weighting scheme that puts emphasis on harder samples. Similarly, hard sample mining Shrivastava et al. (2011) reduces samples in the majority class and selects the most difficult samples to perform training on. In contrast to these works, our weighting mechanism exploits the similarities between classes and ensures noisy samples do not play a significant part in model training.

Supervised contrastive learning is an approach that brings the latent representations of input samples closer together if they belong to the same class (*positives*) and further apart if they belong to different classes (*negatives*). Gunel et al. (2020) use a supervised contrastive loss to improve fine-tuning performance of pre-trained language models in several few-shot learning scenarios. Khosla et al. (2020) introduce a variation of the traditional contrastive loss which aims to produce more samples in the *positive* set. Instead of only considering samples with the same class as belonging to the positive set, they propose to use data augmentation to generate more positive samples. Suresh & Ong (2021) build upon this approach but argue that not all negative samples are equal. To this end, they propose Label-aware Contrastive Loss (LCL) that learns a weight network to infer the relations between classes and weigh samples differently. In contrast, our LANE approach exploits *label-aware margins* to improve the robustness under label noise.

## 3 PROPOSED APPROACH

Here, we first provide background on Area Under the Margin introduced by Pleiss et al. (2020) (§3.1) and then present **L**abel-**A**ware **N**oise **E**limination (LANE), our new approach that improves model robustness in the face of label noise (§3.2).

### 3.1 BACKGROUND

Area Under the Margin (AUM) Pleiss et al. (2020) is a well-established approach that monitors the training dynamics of examples by analyzing their margins during training epochs to automatically identify and remove mislabeled examples from the training data. At training epoch $t$, the margin M Pleiss et al. (2020); Bartlett et al. (2017); Elsayed et al. (2018); Jiang et al. (2018) of an example $\mathbf{x}$ with assigned label $y$ is defined as follows:

$$\mathbf{M}^{(t)}(\mathbf{x}, y) = z_y^{(t)}(\mathbf{x}) - max_{i!=y}z_i^{(t)}(\mathbf{x}) \qquad (1)$$

where $z_y^{(t)}(\mathbf{x})$ is the logit corresponding to assigned label $y$, and $max_{i!=y}z_i^{(t)}(\mathbf{x})$ is the largest *other* logit corresponding to label $i$ (from among all non-assigned labels). The margin measures how different the assigned label is compared to a model's *belief* in a label at some epoch. A negative margin likely implies an incorrect prediction, whereas a positive margin implies a correct prediction. The contribution to generalization of an example $\mathbf{x}$ is measured by averaging the margins of $\mathbf{x}$ across all training epochs T which represents the Area Under the Margin (AUM) Pleiss et al. (2020), defined as follows:

$$\text{AUM}(\mathbf{x}, y) = \frac{1}{\text{T}} \sum_{t=1}^{\text{T}} \mathbf{M}^{(t)}(\mathbf{x}, y) \qquad (2)$$

Figure 1 shows the AUMs of two examples (one correctly labeled and another incorrectly labeled) from an emotion dataset. In the first example, *Makes me sad how brain damage affects boxers*

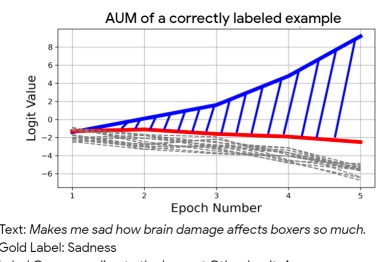 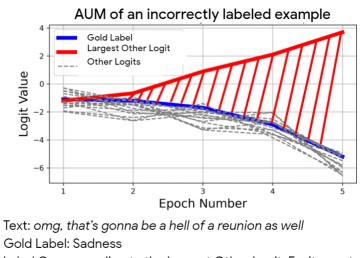

Text: *Makes me sad how brain damage affects boxers so much.*
Gold Label: Sadness
Label Corresponding to the Largest Other Logit: Anger

Text: *omg, that's gonna be a hell of a reunion as well*
Gold Label: Sadness
Label Corresponding to the Largest Other Logit: Excitement

Figure 1: Comparison between the AUM of a correctly labeled example and a mislabeled example.

*so much*, its assigned (or gold) label is "sadness" which is correct and we observe how the logit corresponding to the assigned label grows larger in each epoch, resulting in a positive high AUM. In contrast, in the second example *omg, that's gonna be a hell of a reunion as well*, the assigned (gold) label is "sadness", which is unarguably incorrect, and we observe how other logits, such as the logit corresponding to the "excitement" emotion, are consistently larger than the logit of the "sadness" emotion since the model learns through generalization (from other training examples) that this example shares characteristics of the "excitement" class. Consequently, this example has a low AUM, indicating that its assigned (gold) label is noisy.

Pleiss et al. (2020) first identify mislabeled samples by learning a threshold of separation between the AUMs of clean and erroneous samples through a new artificial class that mimics the training dynamics of mislabeled data and then remove all samples that fall under this threshold. We identified two limitations of the AUM approach. First, we observed (through manual inspection) that through this fixed threshold elimination, difficult but valuable samples that fall under the threshold are unnecessarily removed, and hence, the model has access to less diverse and challenging samples. Second, the current formulation of AUM considers a uniform penalty for each mislabeled sample, irrespective of the semantic similarity between fine-grained classes. A mislabeled example should have a larger negative margin when the wrongly assigned label is more distant from the (hidden) true label and a smaller negative margin when the wrongly assigned label is closer to the (hidden) true label. For example, a sample expressing "excitement" (hidden true label) should have a larger negative margin if the sample is wrongly annotated as "sadness" and a smaller negative margin if the sample is wrongly annotated as "joy". Thus, we argue that the margin M should take into account the inter-class semantic similarities and incur a higher penalty for semantically distant classes and a lower penalty for closely related classes.

### 3.2 Our Proposal: Label-Aware Noise Elimination

We now introduce LANE, our new approach that addresses the above limitations and improves model robustness on fine-grained text classification under label noise. In our approach we redefine the concept of margin to *label-aware margin* to account for the inter-class semantic similarities. Moreover, instead of unnecessarily removing difficult but valuable samples from the training set if they fall under the fixed AUM threshold, we use all samples from the training data, however weighted according to their label-aware margins to reflect inter-class semantic similarities.

**Label-aware Margin (LM)**    Let $\theta$ be a classifier that is trained to predict a task (e.g., sentiment analysis) and $\Pi$ be a weighting network that learns the semantic similarities between classes. To leverage the inherent semantic similarities between classes for dynamic penalty estimation when the assigned label and the prediction do not match we learn a soft-assignment of input samples into all available classes $C$ that accounts for inter-class semantic similarities. Concretely, $\Pi$ optimizes the following label-aware supervised contrastive loss (learned jointly with our classifier $\theta$):

$$\mathcal{L}_{LSCL} = \sum_{\mathbf{x} \in B} \frac{-1}{|P_\mathbf{x}|} \sum_{p \in P_\mathbf{x}} \log \frac{w_{\mathbf{x}, y_\mathbf{x}} \cdot exp(h_\mathbf{x} \cdot h_p)}{\sum_{k \in B \setminus \{\mathbf{x}\}} w_{\mathbf{x}, y_k} \cdot exp(h_\mathbf{x} \cdot h_k)} \tag{3}$$

where $B$ is the current batch, $P_\mathbf{x}$ is the set of positives $p$ for example $\mathbf{x}$ (i.e., in the context of supervised contrastive learning the positives are all examples that belong to the same class as $\mathbf{x}$ and its augmentation Gunel et al. (2020); Khosla et al. (2020)). $h_\mathbf{x}$ is the embedding of $\mathbf{x}$ produced by our model $\theta$. $w_{\mathbf{x}, y_\mathbf{x}}$ and $w_{\mathbf{x}, y_k}$ represent the soft-assignment of example $\mathbf{x}$ to its assigned label $y_\mathbf{x}$ and all

| TEXT | | SDN | JOY | FER | ANG | SRP | DSG | TRS | ANT | M | LM |
|---|---|---|---|---|---|---|---|---|---|---|---|
| | | | | LOGITS | | | | | | | |
| $\mathbf{x}_1$ | The doctors do not have any options for him. | 1.1 | 0.45 | **1.2** | **1.8** | 0.27 | 1.56 | 0.11 | −0.7 | −0.6 | −0.67 |
| $\mathbf{x}_2$ | I have found so much info and support on this site, and yet they accept me for who I am. | 1.1 | 1.56 | **1.2** | 0.45 | 0.27 | 0.11 | **1.8** | −0.7 | −0.6 | −1.15 |

Table 1: Comparison of Margin (M) and Label-aware Margin (LM) for two examples. The assigned label (fear) is shown in **red bold** and the model predicted label for each example is shown in **blue bold**. For both examples, we observe that M is $-0.6$ (i.e., $1.2 - 1.8$). In the first example, LM is rescaled slightly since the assigned emotion fear is semantically close to the emotion corresponding to the largest other logit (i.e., anger). In contrast, we observe that in the second example, the assigned emotion fear is semantically distant from the emotion corresponding to the largest other logit which is trust, and hence, LM becomes much smaller.

the other non-assigned labels $y_k$ where $k = 1, ..., C; k \neq y_{\mathbf{x}}$. To obtain these soft-assignments we utilize the weighting network $\Pi$ applied on top of our model, where $\Pi$ can be viewed as a regular linear layer that projects $h_{\mathbf{x}}$ into a vector $\pi_{\mathbf{x}}$ of length $C$, $\pi_{\mathbf{x}} = \Pi(h_{\mathbf{x}})$. Concretely, $w_{\mathbf{x},y} = \frac{exp(\pi_{\mathbf{x},y})}{\sum_{i=1}^{C} exp(\pi_{\mathbf{x},i})}$.

Using these weights, we propose to rescale the margin and introduce the Label-aware Margin (LM):

$$\text{LM}^{(t)}(\mathbf{x}, y) = \begin{cases} \frac{1}{w_{\mathbf{x},j}} \cdot \mathbf{M}^{(t)}(\mathbf{x}, y) & \text{if} \quad \mathbf{M}^{(t)}(\mathbf{x}, y) < 0 \text{ and } j = \text{argmax}_{i!=y} z_i^{(t)}(\mathbf{x}) \\ \mathbf{M}^{(t)}(\mathbf{x}, y) & \text{otherwise} \end{cases}$$

(4)

where $w_{\mathbf{x},j}$ is the weight obtained using the weighting network $\Pi$, which produces higher values if the (potentially wrong) assigned label $y$ of $\mathbf{x}$ is semantically close to the (hidden) true label $j$ predicted by the model, and lower values otherwise (i.e., if the potentially wrong assigned label is semantically distant from the model prediction). Note that we scale the margins only if the margins are negative, since these are the potentially problematic examples that may be overly ambiguous or mislabeled. To showcase the difference between our proposed label-aware margin LM and the vanilla margin M, we show in Table 1 two examples from an emotion dataset alongside the logits produced by the model as well as the margin M and label-aware margin LM. Both of these examples have the assigned label the *fear* emotion—while $\mathbf{x}_1$ can be viewed as ambiguous, $\mathbf{x}_2$ is clearly mislabeled. However, although the margin of both examples is the same M $= -0.6$, we notice that the assigned label fear is semantically close to the label corresponding to the largest other logit (i.e., anger)—the model prediction in the first example, whereas in the second example, it is semantically distant from the label corresponding to the largest other logit (i.e., trust)—the model prediction. We emphasize that our LM captures this semantic difference between labels. Specifically, we observe that the LM of the first example, where the prediction and the assigned label are semantically close, i.e., anger and fear, is larger than the LM of the second example where the prediction and the assigned label are semantically distant, i.e., trust and fear.

**Average Label-aware Margin (ALM)**  At an arbitrary iteration $t$ we measure the contribution of training examples to learning and generalization by averaging the LMs across the training process, from the beginning up until the current iteration $t$ and obtain the Average Label-aware Margin (ALM) as follows: $\text{ALM}^{(t)}(\mathbf{x}, y) = \frac{1}{t} \sum_{r=1}^{t} \mathbf{M}^{(r)}(\mathbf{x}, y)$.

**Mitigating the harmful effect of mislabed examples**  To mitigate the harmful effect of mislabeled or noisy examples, we propose to use a weighted cross entropy loss during training and assign higher weights for high-ALM examples and lower weights otherwise. Let $N^t = \{\mathbf{x}_i \mid \text{ALM}^{(t)}(\mathbf{x}_i, y_i) < 0\}$ be the set of examples that have negative ALMs up until training iteration $t$ and $\text{ALM}(N^t)$ be the distribution of their ALMs. At $t$, we propose to scale down the loss on examples from $N^t$ for those examples whose ALM is below the mean of the ALM distribution $\text{ALM}(N^t)$. Specifically, we propose to dynamically fit a truncated Gaussian distribution of mean $\mu_t$ and variance $\sigma_t$ at training iteration $t$. We assign a weight for each example $\mathbf{x}_i$ at iteration $t$ as follows:

$$\lambda_{CE}^t(\mathbf{x}_i, y_i) = \begin{cases} \exp(-\frac{(\text{ALM}^{(t)}(\mathbf{x}_i, y_i) - \mu_t)^2}{2\sigma_t^2}) & \text{if } \mathbf{x}_i \in N^t \text{ and } \text{ALM}^t(\mathbf{x}_i, y_i) < \mu_t \\ 1 & \text{otherwise} \end{cases}$$

(5)

During training, we estimate the mean $\mu_t$ and variance $\sigma_t$ using the historical predictions of the model:

$$\mu_t = \frac{1}{|N^t|} \sum_{(\mathbf{x},y) \in N^t} \text{ALM}^{(t)}(\mathbf{x}, y) \tag{6}$$

$$\sigma_t = \frac{1}{|N^t|} \sum_{(\mathbf{x},y) \in N^t} (\text{ALM}^{(t)}(\mathbf{x}, y) - \mu_t)^2 \tag{7}$$

Intuitively, a low weight for an example indicates that the example produced an ALM that is consistently below the mean of the negative ALM distribution. As we have shown, such examples are potentially mislabeled and may hurt generalization. To mitigate this effect, at each training iteration $t$ we simply rescale the cross entropy loss, assigning lower weight to potentially mislabeled examples:

$$\mathcal{L}_{CE} = \sum_{i=1}^{|B|} \lambda_{CE}^t(\mathbf{x}_i, y_i) \cdot H(\theta(\mathbf{x}_i), y_i) \tag{8}$$

where $\theta(\mathbf{x})$ is the probability ditribution of the model $\theta$ on example $\mathbf{x}$, $|B|$ is the batch size, and $H$ is the cross-entropy.

The final loss in LANE is a combination of the weighted cross entropy loss and the contrastive loss:

$$\mathcal{L} = \alpha \cdot \mathcal{L}_{CE} + (1 - \alpha) \cdot \mathcal{L}_{LSCL} \tag{9}$$

In our experiments we set $\alpha = 0.5$.

## 4 EXPERIMENTS

### 4.1 LABEL NOISE

We evaluate the effectiveness of LANE on ten datasets under various amounts of label noise. We employ three setups: **1)** Original datasets, where the label noise comes from annotation errors in the dataset collection process, **2)** 20% noise, where we randomly shuffle the labels of 20% of the training data, and **3)** 40% noise, where we perform the same process for 40% of the training examples.

### 4.2 EXPERIMENTAL SETUP

We carry out all our experiments using an Nvidia A5000 GPU. We use the HuggingFace Transformers Wolf et al. (2020) library for our BERT implementation. The datasets we consider make their train/validation/test splits available, hence, we use the provided splits in our experiments. Similar to Khosla et al. (2020), to expand the positive set of examples in the contrastive loss, we augment our data using synonym replacement Kolomiyets et al. (2011), SwitchOut Wang et al. (2018), and backtranslation Tiedemann & Thottingal (2020). In backtranslation we translate from English to German and back to English. For each batch, we generate 7 augmentations. For all datasets we follow the evaluation metrics used in the works introducing the datasets. The initial batch size is set to 32, hence the total batch size (i.e., including augmentations) is 256. In our training setup, we only scale down the importance of examples during training if their ALM is below a threshold that we set as the ALM mean of examples with negative ALMs (Eq. 6). We also experimented with different ALM thresholds such as 0, but observed slightly worse performance than using the mean.

### 4.3 DATASETS

The datasets used to evaluate LANE are: **1. Empathetic Dialogues** Rashkin et al. (2019), a dataset composed of conversations between a speaker and a listener annotated with 32 emotions. We consider solely the first turn of the conversation in our experiments, resulting in $22,000$ total examples. **2. GoEmotions** Demszky et al. (2020), a sentence-level dataset created using Reddit comments that contains more than $58,000$ sentences annotated with 27 emotions. **3. ISEAR** (International Survey on Emotion Antecedents and Reactions) Scherer & Wallbott (1994), a dataset of $7,700$ personal

experiences annotated with 7 emotions. **4. CancerEMO** Sosea & Caragea (2020), a dataset of $8,500$ examples collected from a cancer forum annotated at sentence level with the 8 basic Plutchik-8 Plutchik (1980) emotions. **5. RCV1** Lewis et al. (2004), a large scale dataset composed of news stories labeled with a total of 105 different topics. **6. SciHTC** Sadat & Caragea (2022), a dataset from $186,160$ scientific papers, annotated with 80 possible topics, **7. SST5** Socher et al. (2013b), a dataset composed of $11,855$ sentences from movie reviews, annotated with five sentiment labels: *negative*, *somewhat negative*, *neutral*, *somewhat positive*, and *positive*. **8. Amazon Review** McAuley & Leskovec (2013), a sentiment classification dataset composed of $600,000$ training and $130,000$ test Amazon reviews annotated with 5 sentiment classes. **9. Yelp Review** Asghar (2016), a sentiment classification dataset with $130,000$ training and $10,000$ test samples annotated with the same 5 classes, and **10. Yahoo Answer** Chang et al. (2008), a topic classification dataset with 10 topic classes, composed of $140,000$ training and $6,000$ test samples.

## 4.4 BASELINE MODELS

We use BERT Devlin et al. (2019) base uncased model in all experiments (denoted by BASE). We compare LANE against methods that use training dynamics to assess the data quality, as well as approaches focused on exploiting the relationships between classes and approaches aimed at learning under label noise:

**Data Cartography** Following Swayamdipta et al. (2020), we identify three types of training examples: easy-to-learn (E2L), hard-to-learn (H2L), and ambiguous (AMG) and analyze the importance of each type to the training process by removing the other two types.

**Noise Layer** Following Goldberger & Ben-Reuven (2016), we introduce a noise layer to the BERT model which we train for correct label estimation. We denote this model by NSE in our experiments.

**Peer Loss Function** We also compare our method against Peer Loss Function (PLF) Liu & Guo (2020), a method that alters the training loss function to account for label noise.

**Area Under the Margin** We consider the AUM method Pleiss et al. (2020) as one of our baselines. This method computes Area Under the Margin metric for each training example and eliminates low-AUM examples that are potentially noisy, using a fixed threshhold for elimination.

**Contrastive Learning:** We compare LANE to the label-aware supervised contrastive learning (LCL) method proposed by Suresh & Ong (2021) and the traditional supervised contrastive learning (SCL) Khosla et al. (2020).

**DISC** Li et al. (2023) proposes an instance-specific dynamic thresholding mechanism that blocks access to specific training examples based on the momentum of each instance's memorization strength. Additionally, DISC proposes to correct the labels of potentially noisy examples.

**UNICON** Karim et al. (2022) leverages semi-supervised learning (SSL) to mitigate the harmful effects of noisy labels by considering the potentially noisy labeled data as unlabeled examples in an SSL algorithm. UNICON also proposes a new selection mechanism for these unlabeled examples during training.

## 5 RESULTS

**Results on Original Datasets**  We show the results on our datasets in Table 2. We make the following observations. **LANE outperforms the baselines in all setups**. We observe improvements of $1.6\%$ weighted F1 on ISEAR, $1.4\%$ weighted F1 on RCV1, $1.5\%$ accuracy on Amazon Review and $1.3\%$ accuracy on Yahoo over the best performing baseline. Notably, over the base BERT model, we see a $2.9\%$ weighted F1 improvement on GoEmotions and $3\%$ improvement on Yahoo. We note that LCL, which leverages inter-class relations through the label-aware contrastive learning loss is the best performing baseline in 5 out of the 10 datasets. Since LANE utilizes similar inter-class relations during training, we postulate improvements over LCL arise from correctly identifying mislabeled or ambiguous examples and eliminating their harmful effect during training.

**Results on 20% Noise Datasets** The results obtained on the $20\%$ noise (20N) datasets where $20\%$ of the labels are intentionally flipped are shown in Table 3. We observe that this setup is significantly more challenging for the model. For instance, on Empathetic Dialogues the weighted F1 of the BASE

| Dataset | Empathetic Dialogues (wF1) | GoEmotions (wF1) | ISEAR (wF1) | CancerEmo (wF1) | RCV1 (wF1) |
|---|---|---|---|---|---|
| BASE | $58.5 \pm 1.2$ | $63.6 \pm 1.2$ | $71.5 \pm 0.6$ | $75.8 \pm 0.8$ | $56.8 \pm 0.8$ |
| E2L | $57.6 \pm 0.8$ | $63.2 \pm 1.2$ | $71.3 \pm 0.7$ | $75.9 \pm 0.9$ | $54.3 \pm 1.1$ |
| H2L | $58.9 \pm 1.4$ | $64.2 \pm 0.7$ | $72.0 \pm 0.6$ | $76.3 \pm 1.3$ | $55.8 \pm 1.4$ |
| AMG | $59.0 \pm 0.6$ | $\underline{64.8 \pm 0.6}$ | $\underline{73.4 \pm 0.5}$ | $76.1 \pm 0.8$ | $52.3 \pm 1.1$ |
| NSE | $58.1 \pm 1.9$ | $63.8 \pm 1.1$ | $72.2 \pm 0.8$ | $76.2 \pm 0.7$ | $55.7 \pm 1.3$ |
| PLF | $58.4 \pm 1.1$ | $63.4 \pm 0.8$ | $71.9 \pm 1.2$ | $75.9 \pm 0.6$ | $56.7 \pm 2.2$ |
| AUM | $58.4 \pm 0.6$ | $63.1 \pm 1.3$ | $71.8 \pm 0.8$ | $76.0 \pm 0.9$ | $56.3 \pm 0.6$ |
| LCL | $59.1 \pm 1.0$ | $\underline{64.8 \pm 0.7}$ | $72.4 \pm 0.5$ | $76.5 \pm 0.9$ | $\underline{57.9 \pm 0.6}$ |
| SCL | $58.9 \pm 0.7$ | $62.8 \pm 1.1$ | $71.5 \pm 0.9$ | $76.2 \pm 0.6$ | $56.9 \pm 1.7$ |
| DISC | $\underline{59.4 \pm 0.9}$ | $63.2 \pm 1.4$ | $72.3 \pm 1.3$ | $76.4 \pm 1.1$ | $56.5 \pm 1.4$ |
| UNICON | $58.4 \pm 0.7$ | $63.1 \pm 0.9$ | $72.5 \pm 1.1$ | $\underline{76.6 \pm 1.3}$ | $56.9 \pm 1.1$ |
| LANE | $\mathbf{60.8 \pm 0.9}$ | $\mathbf{66.5 \pm 0.5}$ | $\mathbf{74.3 \pm 0.4}$ | $\mathbf{78.2 \pm 0.7}$ | $\mathbf{59.3 \pm 0.9}$ |
| DATASET | SciHTC (MF1) | SST-5 (Acc) | Amazon Review (Acc) | Yelp (Acc) | Yahoo (Acc) |
| BASE | $32.5 \pm 1.75$ | $56.3 \pm 0.6$ | $67.5 \pm 0.6$ | $65.9 \pm 0.6$ | $75.4 \pm 0.6$ |
| E2L | $31.6 \pm 1.5$ | $55.7 \pm 1.1$ | $62.9 \pm 0.9$ | $62.8 \pm 2.3$ | $70.4 \pm 1.5$ |
| H2L | $32.2 \pm 1.1$ | $56.6 \pm 1.4$ | $67.9 \pm 0.8$ | $62.3 \pm 1.7$ | $74.1 \pm 1.8$ |
| AMG | $30.6 \pm 1.1$ | $55.1 \pm 1.3$ | $67.4 \pm 1.1$ | $65.1 \pm 1.5$ | $72.3 \pm 1.7$ |
| NSE | $32.8 \pm 1.5$ | $54.1 \pm 1.1$ | $65.8 \pm 1.7$ | $65.1 \pm 1.3$ | $74.6 \pm 1.1$ |
| PLF | $32.2 \pm 1.4$ | $55.7 \pm 1.1$ | $67.4 \pm 2.1$ | $65.8 \pm 1.8$ | $74.8 \pm 1.6$ |
| AUM | $31.2 \pm 2.63$ | $56.4 \pm 0.9$ | $66.4 \pm 0.6$ | $\underline{68.1 \pm 0.6}$ | $72.9 \pm 0.6$ |
| LCL | $\underline{33.1 \pm 1.42}$ | $\underline{57.6 \pm 0.9}$ | $\underline{68.2 \pm 0.6}$ | $66.8 \pm 0.6$ | $76.8 \pm 0.6$ |
| SCL | $32.7 \pm 1.1$ | $56.8 \pm 1.5$ | $67.8 \pm 1.3$ | $66.1 \pm 1.7$ | $75.3 \pm 1.1$ |
| DISC | $32.8 \pm 1.5$ | $56.7 \pm 1.3$ | $67.8 \pm 2.4$ | $66.4 \pm 2.2$ | $75.1 \pm 1.7$ |
| UNICON | $32.7 \pm 1.1$ | $56.5 \pm 1.6$ | $67.5 \pm 1.4$ | $67.9 \pm 1.3$ | $\underline{77.1 \pm 1.5}$ |
| LANE | $\mathbf{34.1 \pm 0.87}$ | $\mathbf{58.9 \pm 0.4}$ | $\mathbf{69.7 \pm 0.6}$ | $\mathbf{69.2 \pm 0.6}$ | $\mathbf{78.4 \pm 0.6}$ |

Table 2: Results of LANE on the fine-grained text classification datasets. The reported results are averaged across five runs and standard deviations are provided. Best results are shown in **bold blue** and second best are underlined.

| Dataset | Empathetic Dialogues (wF1) | GoEmotions (wF1) | ISEAR (wF1) | CancerEmo (wF1) | RCV1 (wF1) |
|---|---|---|---|---|---|
| BASE | $11.6 \pm 3.4$ | $21.5 \pm 2.8$ | $37.6 \pm 3.0$ | $46.7 \pm 1.9$ | $44.4 \pm 3.8$ |
| E2L | $10.3 \pm 0.8$ | $22.6 \pm 1.2$ | $37.1 \pm 0.7$ | $47.5 \pm 0.9$ | $44.3 \pm 1.5$ |
| H2L | $10.6 \pm 1.4$ | $21.8 \pm 0.7$ | $37.3 \pm 0.6$ | $47.9 \pm 1.3$ | $45.8 \pm 2.4$ |
| AMG | $11.4 \pm 1.2$ | $22.1 \pm 0.6$ | $36.9 \pm 0.5$ | $48.4 \pm 0.8$ | $45.9 \pm 2.7$ |
| NSE | $10.2 \pm 1.9$ | $15.6 \pm 1.1$ | $36.4 \pm 0.8$ | $44.2 \pm 0.7$ | $44.9 \pm 1.8$ |
| AUM | $\underline{14.5 \pm 0.6}$ | $\underline{23.5 \pm 1.3}$ | $38.6 \pm 0.8$ | $49.8 \pm 0.9$ | $47.6 \pm 2.7$ |
| SCL | $10.4 \pm 1.4$ | $21.4 \pm 1.3$ | $37.3 \pm 0.9$ | $46.4 \pm 1.1$ | $45.2 \pm 1.5$ |
| LCL | $10.8 \pm 3.24$ | $22.1 \pm 5.1$ | $38.3 \pm 1.5$ | $46.6 \pm 1.2$ | $47.2 \pm 2.2$ |
| DISC | $11.3 \pm 1.0$ | $22.5 \pm 0.7$ | $\mathbf{40.5 \pm 0.5}$ | $\underline{50.3 \pm 0.9}$ | $47.1 \pm 2.2$ |
| UNICON | $10.4 \pm 1.4$ | $21.9 \pm 1.2$ | $39.5 \pm 0.9$ | $42.3 \pm 0.9$ | $\underline{49.2 \pm 2.3}$ |
| LANE | $\mathbf{15.9 \pm 1.3}$ | $\mathbf{24.3 \pm 1.2}$ | $\underline{40.4 \pm 0.8}$ | $\mathbf{52.5 \pm 0.9}$ | $\mathbf{49.4 \pm 2.1}$ |
| DATASET | SciHTC (MF1) | SST-5 (Acc) | Amazon Review (Acc) | Yelp (Acc) | Yahoo (Acc) |
| BASE | $24.5 \pm 4.6$ | $48.9 \pm 3.7$ | $61.5 \pm 1.5$ | $60.7 \pm 1.3$ | $64.8 \pm 1.7$ |
| E2L | $24.1 \pm 2.4$ | $48.2 \pm 2.7$ | $60.7 \pm 2.4$ | $62.3 \pm 2.9$ | $64.9 \pm 3.1$ |
| H2L | $26.7 \pm 2.3$ | $48.7 \pm 1.9$ | $60.9 \pm 2.3$ | $62.6 \pm 2.1$ | $65.7 \pm 1.8$ |
| AMG | $26.9 \pm 1.4$ | $49.4 \pm 1.5$ | $61.3 \pm 2.4$ | $62.9 \pm 2.3$ | $66.5 \pm 1.8$ |
| NSE | $26.7 \pm 4.3$ | $50.4 \pm 4.1$ | $61.7 \pm 3.5$ | $63.5 \pm 3.3$ | $67.2 \pm 2.5$ |
| AUM | $27.4 \pm 4.2$ | $50.4 \pm 2.5$ | $\underline{62.4 \pm 1.7}$ | $63.3 \pm 1.4$ | $65.9 \pm 2.4$ |
| LCL | $24.2 \pm 3.9$ | $48.5 \pm 5.7$ | $61.7 \pm 2.4$ | $63.1 \pm 3.1$ | $65.9 \pm 3.0$ |
| SCL | $24.1 \pm 3.4$ | $51.5 \pm 3.2$ | $62.3 \pm 3.5$ | $\underline{63.7 \pm 3.9}$ | $66.8 \pm 2.5$ |
| DISC | $27.5 \pm 2.1$ | $\underline{51.7 \pm 2.6}$ | $62.1 \pm 2.7$ | $\underline{63.2 \pm 2.5}$ | $\underline{67.3 \pm 2.1}$ |
| UNICON | $\underline{28.9 \pm 3.4}$ | $50.8 \pm 3.1$ | $61.5 \pm 3.7$ | $62.3 \pm 3.9$ | $64.2 \pm 3.7$ |
| LANE | $\mathbf{30.5 \pm 2.97}$ | $\mathbf{53.1 \pm 1.6}$ | $\mathbf{63.1 \pm 2.3}$ | $\mathbf{65.2 \pm 3.1}$ | $\mathbf{68.9 \pm 2.5}$ |

Table 3: Performance of LANE on the ten fine-grained classification datasets in $20\%$ noise setting. The reported results are averaged across five runs and standard deviations are provided. Best results are shown in **bold blue** and second best are underlined.

model drops from $58.5\%$ on the original dataset to $11.6\%$ on the 20N dataset, with a similar trend on all the other datasets. However, even in this more challenging setup, LANE still outperforms all baselines in all setups. For example, on SST5, LANE outperforms AUM in accuracy by $2.7\%$, DISC by $1.4\%$, UNICON by $2.3\%$, and SCL by $1.6\%$. The improvements over the base model are larger, with an average performance increase of $4.5\%$.

**Results on 40% Noise Datasets** We show the results in this high-noise setup in Appendix A.

## 6 ANALYSIS

**Ablation Study** Here, we analyze the effectiveness of various components of our method. To this end, we first design a version of LANE that uses averaged margins instead of ALMs so that the semantic relations are not incorporated into the model. We achieve this by replacing the ALM term in Eq. 5 with AUM and denote this method by LANE$^{-sim}$. Second, we investigate the performance of our approach when completely removing the ALM-based weighting. Specifically, we remove the $\lambda$ weight in Eq. 8 (or set it to 1 always) and train our model to optimize the combination of the contrastive loss and the traditional cross-entropy. We denote this second approach by LANE$^{-alm}$. Finally, we compare LANE against the vanilla AUM Pleiss et al. (2020), which completely removes

| DATASET: | Empathetic Dialogues (wF1) | GoEmotions (mF1) | ISEAR (Acc) | CancerEmo (mF1) | RCV1 (mF1) |
|---|---|---|---|---|---|
| LANE$^{-sim}$ | $14.7 \pm 1.1$ | $22.9 \pm 0.4$ | $39.6 \pm 0.5$ | $50.1 \pm 0.8$ | $45.2 \pm 0.8$ |
| LANE$^{-alm}$ | $13.8 \pm 0.9$ | $21.6 \pm 0.5$ | $37.2 \pm 0.8$ | $46.1 \pm 1.4$ | $46.2 \pm 1.4$ |
| AUM | $14.5 \pm 0.6$ | $23.5 \pm 1.3$ | $38.6 \pm 0.8$ | $49.8 \pm 0.9$ | $47.6 \pm 2.7$ |
| LANE | $\mathbf{15.9 \pm 1.3}$ | $\mathbf{24.3 \pm 1.2}$ | $\mathbf{40.4 \pm 0.8}$ | $\mathbf{52.5 \pm 0.9}$ | $\mathbf{49.4 \pm 2.1}$ |

| DATASET | SciHTC (MF1) | SST-5 (Acc) | Amazon Review (Acc) | Yelp (Acc) | Yahoo (Acc) |
|---|---|---|---|---|---|
| LANE$^{-sim}$ | $28.5 \pm 0.8$ | $50.6 \pm 0.8$ | $61.2 \pm 0.8$ | $62.4 \pm 0.8$ | $67.1 \pm 0.8$ |
| LANE$^{-alm}$ | $29.3 \pm 1.2$ | $50.2 \pm 1.2$ | $61.3 \pm 1.2$ | $64.2 \pm 1.2$ | $66.3 \pm 1.2$ |
| AUM | $27.4 \pm 4.2$ | $50.4 \pm 2.5$ | $62.4 \pm 1.7$ | $63.3 \pm 1.4$ | $65.9 \pm 2.4$ |
| LANE | $\mathbf{30.5 \pm 2.97}$ | $\mathbf{53.1 \pm 1.6}$ | $\mathbf{63.1 \pm 2.3}$ | $\mathbf{65.2 \pm 3.1}$ | $\mathbf{68.9 \pm 2.5}$ |

Table 4: Ablation study: comparison between LANE, LANE$^{-sim}$, LANE$^{-alm}$ and vanilla AUM on the datasets using 20% noise. Best results are shown in **bold blue** and second best are underlined.

| DATASET: | Empathetic Dialogues (wF1) | GoEmotions (mF1) | ISEAR (ACC) | CancerEMO (mF1) | RCV1 (mF1) |
|---|---|---|---|---|---|
| CHATGPT | $12.8 \pm 3.1$ | $21.4 \pm 2.5$ | $37.3 \pm 1.1$ | $48.9 \pm 1.9$ | $42.9 \pm 4.6$ |
| LLAMA-2 | $10.9 \pm 3.7$ | $20.4 \pm 2.7$ | $35.4 \pm 1.6$ | $50.2 \pm 1.7$ | $39.7 \pm 1.8$ |
| LANE | $\mathbf{15.9 \pm 1.3}$ | $\mathbf{24.3 \pm 1.2}$ | $\mathbf{40.4 \pm 0.8}$ | $\mathbf{52.5 \pm 0.9}$ | $\mathbf{49.4 \pm 2.1}$ |

| DATASET: | SciHTC (MF1) | SST-5 (Acc) | Amazon Review (Acc) | Yelp (Acc) | Yahoo (Acc) |
|---|---|---|---|---|---|
| CHATGPT | $28.3 \pm 5.0$ | $49.6 \pm 0.6$ | $62.6 \pm 0.9$ | $64.5 \pm 0.9$ | $64.9 \pm 0.9$ |
| LLAMA-2 | $15.1 \pm 5.2$ | $\mathbf{54.2 \pm 0.4}$ | $61.3 \pm 2.3$ | $62.3 \pm 1.4$ | $61.1 \pm 2.3$ |
| LANE | $\mathbf{30.5 \pm 2.97}$ | $53.1 \pm 1.6$ | $\mathbf{63.1 \pm 2.3}$ | $\mathbf{65.2 \pm 3.1}$ | $\mathbf{68.9 \pm 2.5}$ |

Table 5: Performance of LANE on the ten benchmark datasets compared with LLMs. Best results are shown in **bold blue** and second best are underlined.

examples in the training set that have low AUMs. We show the results obtained on 20N datasets in Table 4. We observe that LANE outperforms LANE$^{-sim}$, LANE$^{-alm}$ and AUM in all setups. Notably, we see a large improvement on SST-5, where LANE pushes the accuracy score by 2.5% over LANE$^{-sim}$, by 2.9% over LANE$^{-alm}$ and by 2.6% over AUM. On RCV1, which has a large number of classes, LANE improves the micro F1 score significantly, obtaining 49.4%, a boost of 4.2% over LANE$^{-sim}$, 3.2% over LANE$^{-alm}$ and 1.8% over AUM. These results show that our proposed Average Label-aware Margin and semantics-aware contrastive loss play an important role in the success of LANE. To gain further insights into LANE we show in Appendix B an error analysis of LANE predictions on the 20% noise ISEAR dataset.

**Comparison with Large Language Models** We test our approach against few-shot large language models: ChatGPT and Llama-2 13B Touvron et al. (2023) to compare the robustess to label noise of LANE with that of popular LLMs in 20% noise setup. For all datasets except SciHTC we fit a large number of examples in the prompt and set the number of few-shot examples to 100. We use only 10 few-shot examples for SciHTC since the examples (i.e., paper abstracts) are much longer and exceed the context window. Similar to the original 20% noise setup, 20% of the few-shot examples are purposefully mislabeled. To account for the variance produced by the particular few-shot examples selected, we run ChatGPT 10 times with different few-shot examples in the prompt and report average values. Similarly, we run Llama-2 20 times with different few-shot examples and show results in Table 5. We observe that LANE outperforms the LLMs on all datasets except SST5. Notably, LANE improves upon Llama-2 by 15.4% on SciHTC and by 9.7% on RCV1 and improves the performance over ChatGPT by 3.1% accuracy on ISEAR and 6.5% micro F1 on RCV1. Among the LLMs, ChatGPT obtains the best results, outperforming Llama-2 especially in complex tasks such as RCV1 and SciHTC. Concretely, ChatGPT obtains 28.3% macro F1 on RCV1, a 13.2% improvement over Llama-2.

# 7 CONCLUSION

In this work, we introduced LANE, a new approach that boosts the capabilities of deep learning models when learning under increased label noise. LANE leverages the inter-class semantic similarities and utilizes training dynamics to boost the performance in fine-grained text classification. We tested LANE on ten fine-grained text classification datasets where it obtained improvements in performance over strong baselines and prior works. In the future, we plan to extend our approach to other domains and data types, e.g., image classification and the legal domain. We make our code available to further research in this area.

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

| Dataset | Empathetic Dialogues (wF1) | GoEmotions (wF1) | ISEAR (wF1) | CancerEmo (wF1) | RCV1 (wF1) |
|---|---|---|---|---|---|
| BASE | – | – | – | – | – |
| E2L | – | – | – | – | – |
| H2L | – | – | – | – | – |
| AMG | – | – | – | – | – |
| NSE | – | – | – | – | $31.4 \pm 1.7$ |
| AUM | $10.4 \pm 0.6$ | $17.5 \pm 1.3$ | $27.8 \pm 0.8$ | $41.8 \pm 0.9$ | $32.5 \pm 1.3$ |
| LCL | – | – | – | – | – |
| SCL | – | – | – | – | – |
| DISC | $14.1 \pm 1.7$ | $19.6 \pm 0.7$ | $31.4 \pm 0.5$ | $47.6 \pm 0.9$ | $33.7 \pm 1.5$ |
| UNICON | $13.7 \pm 1.4$ | $17.4 \pm 1.2$ | $33.1 \pm 0.9$ | $46.5 \pm 0.9$ | $34.6 \pm 1.5$ |
| LANE | $14.6 \pm 1.2$ | $20.5 \pm 0.9$ | $35.1 \pm 0.7$ | $50.1 \pm 0.6$ | $38.2 \pm 1.7$ |
| DATASET | SciHTC (MF1) | SST-5 (Acc) | Amazon Review (Acc) | Yelp (Acc) | Yahoo (Acc) |
| BASE | – | – | – | – | – |
| E2L | – | – | – | – | – |
| H2L | – | – | – | – | – |
| AMG | – | – | – | – | – |
| NSE | $14.8 \pm 1.5$ | $41.6 \pm 2.3$ | - | $44.7 \pm 2.6$ | - |
| AUM | $17.2 \pm 1.4$ | $42.6 \pm 1.5$ | $51.4 \pm 1.1$ | $52.6 \pm 1.8$ | $42.7 \pm 1.9$ |
| LCL | – | – | – | – | – |
| SCL | – | – | – | – | – |
| DISC | $18.5 \pm 2.3$ | $43.8 \pm 1.8$ | $52.9 \pm 1.9$ | $53.8 \pm 2.3$ | $44.7 \pm 2.1$ |
| UNICON | $19.6 \pm 1.5$ | $43.1 \pm 1.6$ | $55.2 \pm 1.3$ | $53.9 \pm 1.7$ | $44.7 \pm 2.1$ |
| LANE | $20.5 \pm 1.5$ | $45.7 \pm 1.3$ | $56.8 \pm 2.2$ | $56.2 \pm 2.3$ | $46.3 \pm 2.5$ |

Table 6: Performance of LANE on the the ten benchmark datasets under $40\%$ label noise. The reported results are averaged across five runs and standard deviations are provided. Best results are shown in **bold blue** and second best are underlined. Results marked with $-$ indicate that the model did not converge.

## A    DATASETS WITH $40\%$ LABEL NOISE

We show in Table 6 results on the $40\%$ noise (40N) datasets. Results marked with - indicate that the model did not convege. We notice that LANE stays effective across the ten datasets, and we observe that AUM yields poor results on this dataset with very high amounts of noise, indicating that it may not work in high-noise setups. For example, AUM outperforms DISC by an average of $1.5\%$ on 20N across the datasets whereas DISC outperforms AUM on 40N by a significant $2.9\%$. Critically, LANE outperforms both DISC and AUM on 40N by an average of $2.2\%$ and $6.2\%$, respectively.

## B    ERROR ANALYSIS

To provide additional insights into our method, we show in Figure 2 a confusion matrix of our LANE approach compared with LANE$^{-alm}$ and a base BERT model on the 20N ISEAR dataset. We make a few observations. First, we note that LANE$^{-alm}$ improves the capabilities of the model over the plain BERT to distinguish between closely related emotions. For example, we see that there are significantly fewer prediction errors confusing disgust and anger or sadness and anger. This result aligns with the purpose of the contrastive loss in LANE$^{-alm}$, which tries to produce language representations that are useful for distinguishing between confusable classes such as anger, disgust, and sadness. Interestingly, we notice that while the performance on closely confusable classes improves, the performance of the model on opposite or more dissimilar classes degrades. For instance, we observe that the model predicts significantly more examples with disgust as true label in the joy class. However, our LANE solves this drawback and we note that the confusability between opposite classes is considerably improved, outperforming the base BERT model as well substantially. Thus, the combination of contrastive learning with our label-aware approach for learning under label noise is extremely effective, denoting that the two components are complementary by nature: while LANE$^{-alm}$ improves the capabilities of the model of distinguishing between easily confusable classes, our full LANE model improves on both highly confusable/overlapping classes and distant classes.

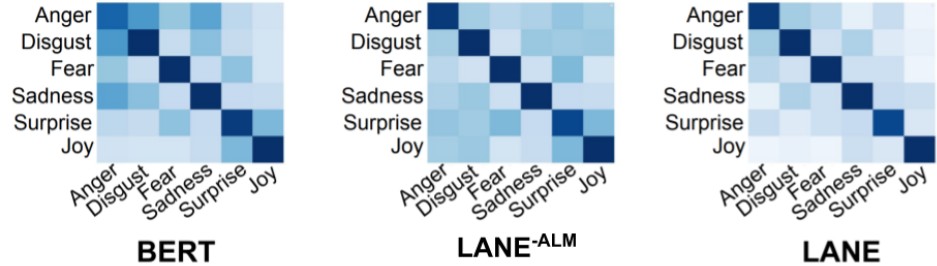

Figure 2: Confusion matrices on the ISEAR dataset created using 20% noise. We compare LANE with a vanilla BERT base model and LANE$^{-alm}$ ablation.

