# OpenReview forum: "LANE: Label-Aware Noise Elimination for Fine-Grained Text Classification"
_ICLR.cc/2025/Conference — Submitted to ICLR 2025_

### Official Review · Reviewer_gSbF · 2024-10-31

**Soundness:** 3
**Presentation:** 2
**Contribution:** 3
**Rating:** 6
**Confidence:** 3

**Summary:**

The paper proposes a new approach, Label-Aware Noise Elimination (LANE), aimed at improving the robustness of deep learning models under label noise in fine-grained text classification. LANE uses semantic relationships between classes and monitors training dynamics to lower the importance of noisy training examples. It is evaluated on various datasets and outperforms baseline methods.

**Strengths:**

The authors address a well-defined problem: the impact of label noise on fine-grained text classification tasks and demonstrates a clear improvement in performance across various datasets with different levels of noise. The idea of leveraging inter-class semantic relationships in noise reduction is an interesting contribution.

**Weaknesses:**

-The overall novelty of this paper is relatively modest, with the main focus being on weighting noisy labeled data.

-Regarding the comparison with LLMs, the study used a 20% noise labeled data setup. However, the correctness of labels in few-shot learning isn't necessarily the key factor in making in-context learning effective. Therefore, introducing noisy labeled data into LLMs' few-shot examples may not hold significant value (refer to the paper Rethinking the Role of Demonstrations: What Makes In-Context Learning Work?).

Overall, I don't believe this is a particularly strong paper, though I may reconsider my score after the rebuttal stage.

**Questions:**

1. Have there been any previous studies on weighting noisy labeled data? Besides using a weighting network for dynamic weighting, were any other weighting approaches explored?
2. How was the hyperparameter α in Equation (9) determined?
3. Why were ChatGPT and Llama2 chosen as the LLM baselines, rather than a more advanced model like GPT-4?

---

> ### Author Response · Authors · 2024-11-25
>
> > The overall novelty of this paper is relatively modest, with the main focus being on weighting noisy labeled data.
>
> We would like to emphasize the contribution of our work below. Many existing datasets contain mislabeled examples. While other prior approaches handle noisy samples in different ways (as we explain below under your question 1), to our knowledge, we are the first to incorporate the degree of “noisiness” of an assigned label (how far away the assigned label is from the true label). Specifically, we assign a higher penalty weight when the (wrongly) assigned label is semantically distant from the (hidden) true label, and lower penalty weight otherwise. We do this through our novel label-aware margin that captures inter-class semantic similarities and can identify mislabeled or noisy samples from the training data. Our results on ten text classification benchmark dataset from different tasks and domains show substantial improvements in performance of LANE compared with strong baselines and prior works, which demonstrates the effectiveness of our approach.
>
> > Although the title mentions "Fine-Grained Text Classification," all ten experimental datasets are centered on sentiment classification, which may call for broader exploration across more diverse scenarios.
>
> Please note that our datasets cover other tasks not just emotion and sentiment classification datasets. Specifically, we also included news stories topic classification (RCV1) with 105 topics/classes, topic classification of research papers (SciHTC) with 80 topics, topic classification of Yahoo question/answer pairs with 10 topic classes (Yahoo Answers).
>
> > Regarding the comparison with LLMs, the study used a 20% noise labeled data setup. However, the correctness of labels in few-shot learning isn't necessarily the key factor in making in-context learning effective. Therefore, introducing noisy labeled data into LLMs' few-shot examples may not hold significant value (refer to the paper Rethinking the Role of Demonstrations: What Makes In-Context Learning Work?).
>
> We performed experiments with LLMs In-Context Learning with the examples in the prompt having the original (gold) labels (i.e., no noise injected) as well as having the 20% injected noise labels. Please see the results with the original (gold) labels in the table below and the 20% injected noise in Table 5 in the paper.
>
> |                                  | ED    | GoEmotions | ISEAR | CancerEmo | RCV1 |
> | -------------------------------- | ----- | ---------- | ----- | --------- | ---- |
> | ChatGPT (original (gold) labels) | 60.3  | 65.3       | 79.3  | 76.3      | 51.4 |
> | Llama2 (original (gold) labels)  | 58.32 | 64.2       | 77.8  | 75.1      | 52.8 |
> | LANE                             | 60.8  | 66.5       | 74.3  | 78.2      | 59.3 |
>
>
> |                                  | SciHTC | SST-5 | Amazon Review | Yelp | Yahoo |
> | -------------------------------- | ------ | ----- | ------------- | ---- | ----- |
> | ChatGPT (original (gold) labels) | 28.7   | 56.9  | 67.3          | 68.1 | 81.4  |
> | Llama2 (original (gold) labels)  | 28.3   | 54.1  | 66.5          | 67.5 | 80.8  |
> | LANE                             | 34.1   | 58.9  | 69.7          | 69.2 | 78.4  |
>
> We found that there is a substantial decrease in performance when we use the 20% noisy labels as compared with original labels. This is in contrast with the paper “Rethinking the Role of Demonstrations: What Makes In-Context Learning Work?”, which is quite interesting and may require a revisit of their analysis with the most recent LLM models, models that learn to follow instructions, etc. The conclusions from this paper could change with the current state of the models and the extremely rapid progress of LLMs. Nevertheless, you make an interesting point and we will include our results and have a discussion on this in our paper.

---

> > ### Author Response · Authors · 2024-11-25
> >
> > > Have there been any previous studies on weighting noisy labeled data? Besides using a weighting network for dynamic weighting, were any other weighting approaches explored?
> >
> > Another study on weighting noisy labeled data that is contemporary with our work is presented in the following paper by Zou et al (2024) https://aclanthology.org/2024.emnlp-main.130.pdf. However, we were not aware of this paper at the time of submission of our paper. We will cite and discuss it in our paper. In this paper, the authors first generate synthetic data and after a required number of samples is obtained, the authors use a cross-model data quality improvement module which re-weights samples using a self-boosting strategy. In contrast, in our work, we utilize all the available training data (without generating synthetic data) and monitor the training dynamics of a model on each training sample to identify potentially mislabeled examples and then re-weight them according to the model’s behavior on these examples measured against their assigned labels. In re-weighting the samples, we estimate the degree of “noisiness” of the assigned labels by introducing label-aware margins averaged across training iterations that capture inter-class semantic similarities. To our knowledge, the estimation of the degree of “noisiness” (how far away the assigned label is from the true label) is novel and unique. Specifically, we assign a higher penalty when the (wrongly) assigned label is semantically distant from the (hidden) true label, and lower penalty otherwise. For example, a sample with true label ``anger'' but with assigned label ``joy'' is noisier (has a higher degree of noisiness) than a sample with true label ``anger'' but with assigned label ``fear'' since ``fear'' is semantically closer to ``anger'' than ``joy''. Our label-aware margin extends the concept of margin by adaptively weighting samples when the (hidden) true label and the (wrongly) assigned label do not match. Precisely, we capture inter-class semantic similarities and dynamically lower samples’ weights if the model perceives them as noisy (the noisier the assigned label the lower the weight). We learn the inter-class semantic similarities using a label-aware supervised contrastive loss. We believe that our label-aware margin is novel and unique and can inspire research on other domains and fields since our results show that our label-aware margin performs much better than the traditional margin. The label-aware margin is one of the major contributions of our work alongside utilizing this margin for re-weighting training samples.
> >
> > Other previous noise-weighting approaches use the model’s confidence in the assigned (gold) label at the current step to assign weights to samples. For example, the work by Jiang et al (2021) “Named Entity Recognition with Small Strongly Labeled and Large Weakly Labeled Data” https://aclanthology.org/2021.acl-long.140.pdf introduces an approach called NEEDLE that uses a model’s confidence in the assigned label at the current iteration to re-weight samples. We compare LANE with the approach by Jiang et al (2021) and show the results in the table below. We observe that LANE performs much better compared with this confidence-based weighting. We will include these results in the paper
> >
> >
> > |                           | ED     | GoEmotions | ISEAR         | CancerEmo | RCV1  |
> > | ------------------------- | ------ | ---------- | ------------- | --------- | ----- |
> > | LANE                      | 60.8   | 66.5       | 74.3          | 78.2      | 59.3  |
> > | NEEDLE (confidence-based) | 51.2   | 61.7       | 70.5          | 74.22     | 54.22 |
> > |                           |        |            |               |           |       |
> > |                           | SciHTC | SST-5      | Amazon Review | Yelp      | Yahoo |
> > | LANE                      | 34.1   | 58.9       | 69.7          | 69.2      | 78.4  |
> > | NEEDLE (confidence-based) | 19.43  | 52.67      | 64.23         | 61.78     | 73.1  |
> >
> > > How was the hyperparameter α in Equation (9) determined?
> >
> > We ran experiments with several values for the hyperparameter α in Equation (9) and found no significant differences in performance. For simplicity then we chose α=0.5.
> >
> > > Why were ChatGPT and Llama2 chosen as the LLM baselines, rather than a more advanced model like GPT-4?
> >
> > No particular reason for choosing ChatGPT and Llama2 other than that we wanted to have results with both commercialized and open-source LLMs. Llama2 performs generally well and ChatGPT is a less-expensive alternative to GPT-4. Nevertheless, we will add experiments with GPT-4 in the paper. We agree it would be interesting to have those results as well.

---

> > > ### Comment · Reviewer_gSbF · 2024-11-25
> > >
> > > Thank you for your submitted response. I have appropriately increased my score. I hope you will continue to refine your paper and provide more details.

---

> > > > ### Author Response · Authors · 2024-11-25
> > > >
> > > > Thank you for reading our response and for your encouragements! We are confident your feedback will help us improve our paper.  Please let us know if there are additional comments and concerns you would like us to address.

---

### Official Review · Reviewer_fNx8 · 2024-11-01

**Soundness:** 3
**Presentation:** 3
**Contribution:** 2
**Rating:** 6
**Confidence:** 2

**Summary:**

In this paper, the authors pointed out the current limitation of area under margin (AUM)-based methods for mitigating label noise. Especially, they suggested the importance of inter-class semantic similarity (e.g., 'joy' is closer to 'happy' than 'fear'). To incorporate this characteristic, they proposed Label-Aware Noise Elimination (LANE) framework. LANE proposes the concept of label-aware margin and methods to mitigate the harmful effect of noisy label data. Their experiment on ten different fine-grained text classification datasets shows the effectiveness of LANE.

**Strengths:**

- The consideration of similarity between labels is interesting and can be meaningful for future studies for fine-grained classification.
- The empirical gain of LANE is considerable compared to other baselines. Furthermore, such gain is consistent, whereas other baselines sometimes have performance degradation.
- It should also be noted that the experiments also include datasets on news and scientific articles, implying that the proposed LANE is not limited to sentiment classification

**Weaknesses:**

I don't think there are any critical weaknesses in this paper. However, I believe that the paper should incorporate more discussion regarding the re-weighting scheme for mitigating noisy labels. For instance, Gao et al. suggested leveraging the concept of bi-level optimization for mitigating noisy labels from synthetic data. Building upon this, Zou et al. proposed a weighting function that does not require computationally expensive bi-level optimization.

**References**
- Gao et al., Self-Guided Noise-Free Data Generation for Efficient Zero-Shot Learning, ICLR 2023.
- Zou et al., FuseGen: PLM Fusion for Data-generation based Zero-shot Learning, arXiv Preprint 2024 (Accepted at EMNLP 2024)

**Questions:**

- Can you present any examples of similar labels in RCV1 and SciHTC datasets?
- Missing reference: Wang et al., Symmetric Cross Entropy for Robust Learning With Noisy Labels, ICCV 2019.
- Please ensure to cite the published version instead of preprints. For instance, "Understanding deep learning requires rethinking generalization" by Zhang et al. was published at ICLR 2017.
- Please use \citet and \citep in correct context.

---

> ### Author Response · Authors · 2024-11-25
>
> Thank you for your insightful comments and feedback. We will incorporate your comments in the paper.
>
> > References Gao et al., Self-Guided Noise-Free Data Generation for Efficient Zero-Shot Learning, ICLR 2023; Zou et al., FuseGen: PLM Fusion for Data-generation based Zero-shot Learning, arXiv Preprint 2024 (Accepted at EMNLP 2024)
>
> Thank you for suggesting these papers. We will cite and discuss them in our paper.
>
> In the paper by Zou et al (2024), the authors first generate synthetic data and after a required number of samples is obtained, the authors use a cross-model data quality improvement module which re-weights samples using a self-boosting strategy. In contrast, in our work, we monitor the training dynamics of the model on each training sample to identify potentially mislabeled samples and then re-weight them according to the model’s behavior on these samples measured against their assigned labels. In re-weighting the samples, we estimate the degree of “noisiness” of the assigned labels (which to our knowledge, is novel and unique to our work) by introducing label-aware margins averaged across training iterations that capture inter-class semantic similarities. Specifically, we assign a higher penalty when the (wrongly) assigned label is semantically distant from the (hidden) true label, and lower penalty otherwise. We believe that our label-aware margin can inspire research on other domains and fields with substantial improvements in performance. The label-aware margin is one of the major contributions of our work alongside utilizing this margin for re-weighting training samples.
>
> > Can you present any examples of similar labels in RCV1 and SciHTC datasets?
>
> We will add examples in the paper with similar labels for RCV1 and SciHTC.
>
> > Missing reference: Wang et al., Symmetric Cross Entropy for Robust Learning With Noisy Labels, ICCV 2019.
>
> We will cite and discuss this missing reference as well.
>
> > Please ensure to cite the published version instead of preprints. For instance, "Understanding deep learning requires rethinking generalization" by Zhang et al. was published at ICLR 2017.
>
> We will cite the published version.
>
> > Please use \citet and \citep in correct context.
>
> We will fix this. Thank you for the suggestions!

---

> > ### Comment · Reviewer_fNx8 · 2024-11-25
> >
> > Thank you for submitting the response. I acknowledged the response and would like to keep the current evaluation.

---

> > > ### Author Response · Authors · 2024-11-25
> > >
> > > Thank you for acknowledging our work and reading our response! We are glad you found that LANE yields significant performance improvements and that our work could positively impact future studies in fine-grained classification. Please let us know if there are additional questions or concerns you would like us to address.

---

### Official Review · Reviewer_ucXh · 2024-11-06

**Soundness:** 2
**Presentation:** 3
**Contribution:** 2
**Rating:** 3
**Confidence:** 4

**Summary:**

This paper presents LANE (Label-Aware Noise Elimination), a method that improves text classification by reducing the impact of noisy labels during training. Using class relationships and training patterns, LANE shows F1-score improvements of 2.4-4.5% across their datasets.
The work's novelty is limited, though. The core idea of weighting noisy samples has been well-explored across NLP and computer vision - from text classification to multi-label tasks, named entity recognition [1], and image segmentation. The evaluation is also restricted to cases with few labels, leaving questions about its effectiveness for problems with hundreds or thousands of classes.
Furthermore, the experimental is limited to BERT-based models, lacking comprehensive ablation studies across different model architectures.

[1] Named Entity Recognition with Small Strongly Labeled and Large Weakly Labeled Data

**Strengths:**

The analysis includes benchmarking against recent generative AI models using few-shot prompting, demonstrating that targeted classification methods remain valuable alongside emerging large language models

**Weaknesses:**

The work's novelty is limited, though. The core idea of weighting noisy samples has been well-explored across NLP and computer vision - from text classification to multi-label tasks, named entity recognition [1], and image segmentation. The evaluation is also restricted to cases with few labels, leaving questions about its effectiveness for problems with hundreds or thousands of classes.
Furthermore, the experimental is limited to BERT-based models, lacking comprehensive ablation studies across different model architectures.

The comparison with generative AI models lacks rigor in prompt optimization - the baseline performance could potentially be improved significantly through systematic prompt engineering or automated tuning frameworks like DSPy. This limitation undermines the fairness of the comparison.

**Questions:**

1. What are the key conceptual differences between LANE and previous noise-weighting methods in text classification and NER? The paper would benefit from a clearer positioning of its technical novelty.
2. The current evaluation focuses on tasks with limited label sets. How would LANE handle extreme classification scenarios with hundreds/thousands of labels, and what adaptations might be needed?
3. The experiments are currently BERT-centric. Could you discuss LANE's potential performance and compatibility with other architectures (T5, RoBERTa)? Is the semantic relationship modeling architecture-independent?
4. The GenAI baselines seem to use basic few-shot prompting. Have you considered using prompt optimization tools like DSPy to establish a more rigorous comparison? How might LANE's advantages compare against such optimized baselines?
5. Have you considered LLM instruction tuning, like tune the LLaMA-2/4 model?

---

> ### Author Response · Authors · 2024-11-25
>
> Thank you for your insightful comments and feedback. We will incorporate your comments in the paper.
>
> > What are the key conceptual differences between LANE and previous noise-weighting methods in text classification and NER? The paper would benefit from a clearer positioning of its technical novelty.
>
> Previous noise-weighting approaches (including the NER approach by Jiang et al (2021) that you pointed out—which is called NEEDLE) use the model’s confidence in the assigned (gold) label at the current step to assign weights to samples. In our paper, we introduce a novel concept that we call label-aware margin that extends the margin (or Area Under the Margin) from machine learning to assign weights to training samples. In re-weighting the samples, we estimate the degree of “noisiness” of the assigned labels by introducing label-aware margins averaged across training iterations that capture inter-class semantic similarities. To our knowledge, the estimation of the degree of “noisiness” (how far away the assigned label is from the true label) is novel and unique. Specifically, we assign a higher penalty when the (wrongly) assigned label is semantically distant from the (hidden) true label, and lower penalty otherwise. For example, a sample with true label ``anger'' but with assigned label ``joy'' is noisier (has a higher degree of noisiness) than a sample with true label ``anger'' but with assigned label ``fear'' since ``fear'' is semantically closer to ``anger'' than ``joy''. Our label-aware margin extends the concept of margin by adaptively weighting samples when the (hidden) true label and the (wrongly) assigned label do not match. Precisely, we capture inter-class semantic similarities and dynamically lower samples’ weights if the model perceives them as noisy (the noisier the assigned label the lower the weight). We learn the inter-class semantic similarities using a label-aware supervised contrastive loss. We believe that our label-aware margin is novel and unique and can inspire research on other domains and fields since our results show that our label-aware margin performs much better than the traditional margin. The label-aware margin is one of the major contributions of our work alongside utilizing this margin for re-weighting training samples. We will clarify this in the paper.
>
> The work by Jiang et al (2021) “Named Entity Recognition with Small Strongly Labeled and Large Weakly Labeled Data” https://aclanthology.org/2021.acl-long.140.pdf that introduced NEEDLE uses a model’s confidence in the assigned label at the current iteration to re-weight samples. We compare LANE with the approach by Jiang et al (2021) and show the results in the table below. We observe that LANE performs much better compared with this confidence-based weighting. We will include these results in the paper.
>
> |                           | ED     | GoEmotions | ISEAR         | CancerEmo | RCV1  |
> | ------------------------- | ------ | ---------- | ------------- | --------- | ----- |
> | LANE                      | 60.8   | 66.5       | 74.3          | 78.2      | 59.3  |
> | NEEDLE (confidence-based) | 51.2   | 61.7       | 70.5          | 74.22     | 54.22 |
> |                           |        |            |               |           |       |
> |                           | SciHTC | SST-5      | Amazon Review | Yelp      | Yahoo |
> | LANE                      | 34.1   | 58.9       | 69.7          | 69.2      | 78.4  |
> | NEEDLE (confidence-based) | 19.43  | 52.67      | 64.23         | 61.78     | 73.1  |

---

> > ### Author Response · Authors · 2024-11-25
> >
> > > The current evaluation focuses on tasks with limited label sets. How would LANE handle extreme classification scenarios with hundreds/thousands of labels, and what adaptations might be needed?
> >
> > Please note that we experimented with datasets that cover a range of labels—from a few labels to 100+ labels. At the same time, LANE will handle extreme classification scenarios with hundreds or thousands of labels seamlessly well and will not need any particular adaptations. It does not matter for the network that learns the inter-class semantic similarities if there are 100 classes or 500 or 1000 classes. The label-aware margin will be calculated in the same way and the overall model will be learned with the same loss which is the combination of the weighted cross entropy loss and supervised contrastive loss. We expect LANE to work well in extreme classification scenarios with hundreds/thousands of labels, especially in the scenarios with confusable classes.
> >
> > > The experiments are currently BERT-centric. Could you discuss LANE's potential performance and compatibility with other architectures (T5, RoBERTa)? Is the semantic relationship modeling architecture-independent?
> >
> > LANE is compatible with other architectures such as T5 and RoBERTa. In our implementation, we just need to change the model from BERT to RoBERTa, and indeed the semantic relationship modeling is architecture-independent. We will include the results of LANE with RoBERTa in our paper and a discussion on this.
> >
> > > The GenAI baselines seem to use basic few-shot prompting. Have you considered using prompt optimization tools like DSPy to establish a more rigorous comparison? How might LANE's advantages compare against such optimized baselines?
> >
> > Our goal with the GenAI baselines was to compare LANE against the widely used few-shot prompting (In-Context Learning) with both open-source and commercialized models. We show that LANE performs better than LLMs with few-shot prompting with substantial improvements. This is common practice/comparison in other papers.
> >
> > > Have you considered LLM instruction tuning, like tune the LLaMA-2/4 model?
> >
> > We did not experiment with LLM fine-tuning since this could be a computationally expensive option. Instead, since LLMs with In-Context Learning (few-shot setting) perform very well on many tasks—often on par or better than fine-tuning, we use this as an approach for comparison in our work.

---

> > > ### Author Response · Authors · 2024-11-25
> > >
> > > Thank you once again for your feedback! Please let us know if our review addresses your comments and concerns. We are happy to answer any additional question or address additional concerns.

---

### Official Review · Reviewer_BfD2 · 2024-11-06

**Soundness:** 3
**Presentation:** 2
**Contribution:** 3
**Rating:** 5
**Confidence:** 3

**Summary:**

This paper addresses the issue of noisy or mislabeled data in single-label text classification. The authors propose an approach that incorporates a label margin, considering semantic similarity between the assigned and predicted labels. This margin is applied to both cross-entropy loss and supervised contrastive loss, forming the training objective. Experimental results indicate that this method outperforms traditional baselines and LLMs with noisy in-context learning on noisy datasets.

**Strengths:**

1. The paper presents clear motivations for the work, specifically identifying limitations of prior methods, such as making samples less challenging and applying uniform penalties, which are well explained.
2. The proposed approach shows strong performance on noisy classification tasks, demonstrating robustness to label noise. The comparisons with existing LLMs add value, though they could be further refined.

**Weaknesses:**

Some essential details are unclear, making it challenging to fully assess the proposed method. While the high-level concept and rationale for the design choices are understandable, additional specifics are needed. For example, in line 208, the authors mention that the weighting network is trained according to Eq. (3), but the weighting network does not appear in this equation, leaving its connection to Eq. (3) ambiguous. Please refer to the questions below for further clarification.

Minor Comment:
The comparison between few-shot LLMs and LANE may be somewhat biased since the LLMs rely only on in-context learning without fine-tuning on the target dataset. To make the comparison fairer, it would be helpful to include LLM performance in both noisy and noiseless scenarios.

**Questions:**

1. How is the weight of the "regular linear layer" in the weighting network obtained, and why is this network expected to predict the semantic similarity between the assigned label and the predicted label? I am unclear on its connection to Eq. (3).
2. Could you provide insights on how this method could be extended to address multi-label classification tasks?
3. In line 255, why is the average label-aware margin summed over M(x,y) instead of LM(x,y)?

---

> ### Author Response · Authors · 2024-11-25
>
> Thank you for your insightful comments and feedback. We will incorporate your comments in the paper.
>
> > The comparison between few-shot LLMs and LANE may be somewhat biased since the LLMs rely only on in-context learning without fine-tuning on the target dataset. To make the comparison fairer, it would be helpful to include LLM performance in both noisy and noiseless scenarios.
>
> Thank you for the suggestion. We experimented with LLMs (ChatGPT and Llama 2) on the noiseless scenario, per your suggestion, and found that LANE’s performance in the noiseless scenario is better than the LLMs’ performance on 8 out of 10 datasets (please see the table below for the results on the noiseless datasets). We will include these results in the paper. These results (plus the ones with 20% noise from Table 5 from the paper) show that LANE outperforms LLMs (ChatGPT and Llama 2) in both scenarios, noiseless and 20% noise.
>
> |         | ED    | GoEmotions | ISEAR | CancerEmo | RCV1 |
> | ------- | ----- | ---------- | ----- | --------- | ---- |
> | ChatGPT | 60.3  | 65.3       | 79.3  | 76.3      | 51.4 |
> | Llama2  | 58.32 | 64.2       | 77.8  | 75.1      | 52.8 |
> | LANE    | 60.8  | 66.5       | 74.3  | 78.2      | 59.3 |
>
> |         | SciHTC | SST-5 | Amazon Review | Yelp | Yahoo |
> | ------- | ------ | ----- | ------------- | ---- | ----- |
> | ChatGPT | 28.7   | 56.9  | 67.3          | 68.1 | 81.4  |
> | Llama2  | 28.3   | 54.1  | 66.5          | 67.5 | 80.8  |
> | LANE    | 34.1   | 58.9  | 69.7          | 69.2 | 78.4  |
>
> > How is the weight of the "regular linear layer" in the weighting network obtained, and why is this network expected to predict the semantic similarity between the assigned label and the predicted label? I am unclear on its connection to Eq. (3).
>
> The architecture of our model is as follows: we have BERT layers that produce the embedding of the input x (i.e., h_x). On top of BERT, we add a fully connected layer (linear layer) that is optimized with the weighted cross-entropy loss (the classifier \theta) and another fully connected layer (linear layer) that is optimized with the label-aware supervised contrastive loss (the weight network \Pi). Both the classifier \theta and the weight network \Pi are learned jointly. Since the weight network is optimized with the supervised contrastive loss, its purpose is to learn a semantic similarity among classes while pushing the examples from the same class close and the examples from different classes further apart. The vector \pi_x of length C (where C is the number of classes) is obtained as \pi_x = \Pi(h_x) (please see lines 228-229 in the paper) and then softmax is used to obtain the weight of each class y as w_{x,y}. This weight (learned through weight network \Pi) is used to rescale the margin of the classifier \theta, which is then integrated in the weighted cross-entropy loss.
>
> Please note that, in principle, we could use only the classifier \theta (without the weight network \Pi) and directly use the probability of each class (after applying softmax) from \theta as weights in the label-aware margin. However, the supervised contrastive loss helps with the closely confusable classes to push them further apart and to learn to better differentiate between the classes by learning a finer semantic similarity among classes.
>
> > Could you provide insights on how this method could be extended to address multi-label classification tasks?
>
> This is a very interesting idea that is worth pursuing in the future. The label-aware margin has to be extended to handle multi-labels to address multi-label classification tasks. The supervised contrastive loss also has to incorporate multiple gold labels per example (which can be done). We leave this for future work as a promising and exciting future direction.
>
> > In line 255, why is the average label-aware margin summed over M(x,y) instead of LM(x,y)?
>
> This is a typo. It should be summed over LM(x,y). Thank you for catching this. We will fix it.

---

> > ### Author Response · Authors · 2024-11-25
> >
> > We would like to thank you again for your insightful comments! Please let us know if our response addresses your concerns and if there are additional questions or clarifications needed. We are happy to address these.

---

### Meta-Review · Area_Chair_2RS8 · 2024-12-20

**Metareview:**

This paper proposes label-aware noise elimination to improve the robustness of deep learning models when trained under increased label noise in fine-grained text classification. This method leverages the semantic relations between classes and monitors training dynamics of the model on each training example to dynamically lower the importance of training examples that are perceived to have noisy labels. Experimental results demonstrate the effectiveness of the proposed method.

Pros:

- The motivation of assigning different weights for different samples to handle label noise is reasonable.
- The performance of the proposed method is good.

Reasons to reject:

- The core idea of weighting noisy samples has well-explored and this paper fails to reflect the key novelty of the proposed method following this widely adopted idea.
- The experiments are currently conducted on BERT-centric model architectures. To demonstrate the generalizability of the proposed method, this paper needs to show the compatibility with other model architectures.

**Additional Comments On Reviewer Discussion:**

This paper finally receives the scores of 6 (Reviewer fNx8), 6 (Reviewer gsbF), 5 (Reviewer BfD2), 3 (Reviewer ucXh). After the rebuttal process, some of the reviewers' concerns have been addressed. I have carefully checked the comments from Reviewer ucXh and agreed with most of the raised concerns. Specifically, The novelty of the proposed method is not high and the experiments are not enough to demonstrate that the proposed method can be compatible with many model architectures.

---

### Decision · Program_Chairs · 2025-01-22

Reject